# Comparing the Effects of Prescribed Burning on Soil Chemical Properties in East Texas Forests with Longleaf and Other Southern Pines in the Overstory

**Cassady P. Dunson, Brian P. Oswald * and Kenneth W. Farrish**

Division of Environmental Science, Arthur Temple College of Forestry and Agriculture,
Stephen F. Austin State University, P.O. Box 6109, Nacogdoches, TX 75962-6109, USA
* Correspondence: boswald@sfasu.edu; Tel.: +1-936-645-7990

**Abstract:** Little has been reported on the effects of repeated prescribed burning on southern United States' forest soils, especially when site preparation is not the prescribed fire objective. This study was aimed at identifying any correlations between the soil chemical properties among differing burn intervals and the effects prescribed burning has on them. Sampling was performed in 36 plots at three sites with two different burn intervals (2–3 years and biannually) and measured properties: (1) pre-burn (before the fire), (2) post-burn (one month after the fire), and (3) at vegetation green-up (three months after the fire). Sites varied by overstory species with longleaf pine (*Pinus palustris*) and shortleaf pine (*Pinus echinata* Mill.) in the overstory on one site, a mix of loblolly pine (*P taeda* L.) and shortleaf pine on another, and longleaf and loblolly pines on the third. SAS was used to determine the effects of prescribed burning between different time spans (pre-burn to post-burn, post-burn to green-up, and pre-burn to green-up) and between the two different burn intervals. We found that there could be short-term responses in soil chemical properties from repeated burning treatments including nitrogen in the forms of ammonium and nitrate, the carbon-to-nitrogen ratio, and electrical conductivity, all of which decreased following fire regardless of overstory species.

**Keywords:** longleaf pine; prescribed burn; fire; forest soils; soil infiltration; soil chemistry

## 1. Introduction

Prescribed burning is utilized in the Western Gulf Coast of East Texas to promote plant diversity, control woody species encroachment, maintain historical disturbance regimes, or achieve wildlife habitat improvement objectives. Because of a lack of quantification of the impact of prescribed burning on soil properties, and the conflicting nature of previous research on this topic, a better understanding of how fire influences soil chemical properties is needed. The limited research that has been performed in southern pine forests often looks at site preparation burning [1], but there is none where fuel reduction or wildlife habitat manipulation are the primary burn objectives.

The intensity and duration of a fire often determine which chemical properties are affected, and the moisture content of fuel sources and fuel loading are major factors when determining the intensity and severity of a fire; these variables may also affect the chemical properties of the soil [2]. Fire may or may not completely consume the organic matter; any residual material may have the ability to migrate down into the subsurface soil and stay for an extended period of time, slowly releasing organic carbon into the soil profile [3,4]. Soils that are severely burned tend to have lower nitrogen levels and higher calcium levels than unburned soils due to volatilization. Also, in a typical prescribed burn, potassium, magnesium, and phosphorous do not usually significantly change after a burn [5–7]; however, a severe fire could cause mass amounts of nutrient volatilization, or nutrients going into the soil may not become bioavailable and remain unused until they dissolve in the soil water and can be taken up by plants [8,9]. Fire can cause nutrients to become

more mobile, either entering into the soil and remaining within the range of plant roots or leaching down into the soil profile and into the ground water [1,7–11]; however, a single low-intensity burn does not usually change the nutrient availability in soils.

Soil pH often increases significantly after a fire in forest soils, possibly caused by an influx of nutrients released from the litter layer by the fire, especially nitrogen mineralization and fixation. Following higher temperature burns, it is typical to see an increase in pH [7,9,10,12] in neutral or slightly acidic soils such as those found in the pine-dominated southeastern United States. The burning of the litter layer is, in most cases, an alkaline reaction that raises the pH as nutrients become mobile and enter the soil profile. Ash may be left behind from the litter layer after a burn often creates an increase in soil pH as well. Slight increases in pH do not last very long after a fire due to the leaching of nutrients below the plant's root zone, runoff from lowered infiltration, or oversaturation of the soil profile [8]. A low-intensity fire regime can, however, lead to an overall increase in soil pH levels [13]. The season of burn also has an effect on the soil nutrients as soil surface temperatures control which nutrients become volatile, which become bioavailable to plants, and which will leach down into the soil profile. Fuel moisture also affects the nutrients as well [6,14], as does fuel loading [15].

The objective of this study was to determine if prescribed burning has an effect on soil chemical parameters in common East Texas forest soils supporting common overstory species, including longleaf pine (*Pinus palustris*).

## 2. Materials and Methods

### 2.1. Study Area

We sampled plots pre-burn, post-burn (one month after the burn), and at green-up (three months after the burn) during the 2020–2021 burn seasons at the USFS Davy Crockett National Forest (DCNF), USFS Angelina National Forest (ANF), and the Winston 8 Land and Cattle Ltd. Tree Farm (Winston 8) in East Texas. Two plots were located on the Davy Crockett National Forest, six plots were located on the Angelina National Forest, and thirty plots on the Winston 8 Land and Cattle Ltd. Tree Farm. A Global Positioning System (GPS) using the Environmental Systems Research Institute (ESRI) ArcGIS mapping system was used to establish plot and subplot locations. Subplot 1 was located at a randomly selected location and direction, not exceeding 15 m from the plot center, and subplot 2 was randomly located a maximum of 50 m from subplot 1 (Table S1).

Winston 8 was predominantly comprised of longleaf pine (*Pinus palustris* Mill.) and scattered shortleaf pine (*P. echinata* Mill.) in the overstory with no notable midstory. The understory was a mixture of wild blackberry (*Rubus* spp. L.), American beautyberry (*Callicarpa americana* L.), and a mixture of grasses (*Poaceae* spp. Barn.), and is burned biannually. The DCNF has a mix of loblolly pine (*P taeda* L.) and shortleaf pine in the overstory, a midstory additionally comprised of hickory (*Carya* spp. Nutt.), and an understory dominated by American beautyberry, wax myrtle (*Myrica* spp. (L.) Small), sassafras (*Sassafras albidum* (Nutt.) Nees), sweetgum (*Liquidambar styraciflua* L.), elm species (*Ulmus* spp. L.), and greenbriar (*Smilax* spp. L.). The ANF differs from the DCNF only in the overstory, which was comprised of longleaf and loblolly pines. Both ANF and DCNF are burned on a 2–3-year interval.

### 2.2. Data Collection

At each subplot, the soil map unit confirmation, O-horizon depth and weight, mineral soil sampling at a depth of 0 to 15 cm, soil organic carbon content, soil nutrients, and soil pH were measured pre-burn, and re-measurements of everything except soil texture and soil series confirmation were taken post-burn and at green-up.

We used the Web Soil Survey [16] developed by the Natural Resources Conservation Service (NRCS) to tentatively identify the map units, and a soil auger was then used for confirmation (Table S2). Mineral soil samples not exceeding 15 cm in depth were taken from each subplot and sent to the Stephen F. Austin State University Soil, Plant, and

Water Analysis Laboratory to determine total organic carbon, total nitrogen, extractable phosphorous, exchangeable potassium, calcium, magnesium, extractable sulfur, extractable sodium, and soil pH. A 22 cm by 22 cm square was placed on the surface of the ground within each subplot, and all of the organic matter (litter and O-horizon) was collected and taken to the lab, weighed, and dried to determine the moisture content. The depth of the O-horizon was also recorded to estimate O-horizon density.

To estimate the soil pH from each subplot sample, we placed 20 g of field moist soil into an Erlenmeyer flask with 40 mL of water and then put it on a shaker for a minimum of 15 min. A pH probe was calibrated using pH standards, inserted into the flask, and used to record the soil pH. We also analyzed the pH in a buffer solution using the Moore–Sikora buffer method. Phosphorus, potassium, calcium, magnesium, and sulfur were extracted by Mehlich-3 extractable nutrients and recorded in mg kg$^{-1}$. Electrical conductivity was reported in μs cm$^{-1}$. Nitrogen was measured as ammonium and nitrate using different wavelengths. Ammonium was measured using a 670 and 800 nm light-wave scale. Nitrate was measured using a 540 and 600 nm wavelength scale.

A two-factor design and the interaction of time and site were constructed using a *t*-test and analyzed at the two burn intervals (and three sampling periods) with a *p*-value of 0.10. We then ran one-way ANOVA tests on each variable to identify specific interaction differences in the burn intervals.

## 3. Results

Soil mapping units were either confirmed or changed to reflect the sampled pedon results. All soil textures were classified as either sand or loamy sand at the 0 to 15 cm depth (Table 1), and it was assumed these soils would likely respond similarly [17].

**Table 1.** Comparisons of soil chemical properties with their mean and change in mean between time frames. Δ = change in mean. * Indicates a significant difference at *p*-value 0.10.

| | Units | Pre-Burn Mean | Post-Burn Mean | Green-Up Mean | Δ Pre-Burn to Post-Burn | Δ Post-Burn to Green-Up | Δ Pre-Burn to Green-Up |
|---|---|---|---|---|---|---|---|
| Phosphorus | mg kg$^{-1}$ | 8.52 | 8.79 | 6.51 | 0.44 | * −2.45 | * −2.01 |
| Potassium | mg kg$^{-1}$ | 45.16 | 54.34 | 41.58 | * 9.18 | * −12.76 | −3.56 |
| Calcium | mg kg$^{-1}$ | 425.14 | 433.36 | 397.77 | 8.22 | −32.98 | −40.91 |
| Magnesium | mg kg$^{-1}$ | 67.10 | 73.98 | 62.62 | * 6.88 | * −11.36 | −4.48 |
| Sulfur | mg kg$^{-1}$ | 4.77 | 4.72 | 3.90 | −0.05 | * −0.81 | * −0.86 |
| Sodium | mg kg$^{-1}$ | 59.43 | 58.93 | 57.03 | −0.50 | −1.89 | * −2.39 |
| pH Water | | 5.03 | 5.01 | 5.48 | −0.01 | * −0.01 | * −0.01 |
| Estimated CEC | cmoles kg$^{-1}$ | 5.42 | 6.09 | 6.26 | 0.75 | 0.06 | * 0.99 |
| Electrical Conductivity | μs cm$^{-1}$ | 73.94 | 76.05 | 41.51 | 2.11 | * −34.51 | * −32.43 |
| Carbon/Nitrogen Ratio | | 20.79 | 21.44 | 19.72 | 0.66 | * −1.72 | −1.07 |
| Organic Matter | % | 3.36 | 3.49 | 3.05 | −0.01 | −0.01 | −0.01 |
| Total Carbon | % | 1.68 | 1.75 | 1.53 | 0.01 | −0.01 | −0.01 |
| Total Nitrogen | % | 0.08 | 0.08 | 0.08 | −0.01 | −0.01 | −0.01 |
| Nitrogen as Ammonium | mg kg$^{-1}$ | 5.03 | 8.30 | 5.08 | * 3.27 | * −3.22 | 0.05 |
| Nitrogen as Nitrate | mg kg$^{-1}$ | 8.24 | 10.16 | 2.11 | 3.91 | * −7.61 | −4.07 |

Phosphorus had a significant decrease over the post-burn to green-up and pre-burn to green-up time frames, and significantly differed across all time frames between the two different burn intervals (Winston 8's biannual and DCNF and ANF's 2–3-year interval). There was also a significant increase in potassium and magnesium over the pre-burn to post-burn time frame and a significant decrease over the post-burn to green-up time frame. Calcium only had a significant difference at green-up between the two burn intervals and increased over the pre-burn to post-burn time frame. Magnesium had a significant increase over the pre-burn to post-burn time frame, but a significant decrease from post-burn to green-up and also had a significant difference between the two burn intervals at green-up.

There was a significant decrease in sulfur from post-burn to green-up and pre-burn to green-up. Sodium significantly decreased from post-burn to green-up and pre-burn to green-up (Table 1, Tables S3 and S4).

The pH analyzed in water significantly decreased post-burn to green-up and pre-burn to green-up. The pH in both water and the buffer solution was used to estimate the cation exchange capacity (CEC), which significantly increased from pre-burn to green-up, and was significantly different between the burn intervals during the post-burn and green-up time frames (Tables 2 and 3). Electrical conductivity significantly decreased from post-burn to green-up and pre-burn to green-up, with an increase over the pre-burn to post-burn interval. There was also a significant difference at post-burn and green-up between the two burn intervals.

The carbon-to-nitrogen ratio significantly decreased from post-burn to green-up, with a significant difference at all time frames between the two burn intervals. The soil organic matter percentage showed a significant difference at pre-burn and post-burn, but the carbon percentage and nitrogen percentage did not have significant statistical differences; however, the total carbon percentage was only significantly different between pre-burn and post-burn, while the total nitrogen percentage was only significantly different post-burn between the two burn intervals. Nitrogen measured as ammonium showed a significant increase from pre-burn to post-burn, followed by a significant decrease over the post-burn to green-up time frame. Nitrogen measured as nitrate significantly decreased from post-burn to green-up (Tables 1, 2 and S4).

**Table 2.** One-way ANOVA of soil chemical properties comparing the differences between two burn intervals at three different time frames. NF = National Forest burn interval; W8 = Winston 8 burn interval. * Indicates significant difference at *p*-value 0.10.

| Variable | Units | Pre-Burn NF Mean | Pre-Burn W8 Mean | Post-Burn NF Mean | Post-Burn W8 Mean | Green-Up NF Mean | Green-Up W8 Mean | Pre-Burn *p*-Value | Post-Burn *p*-Value | Green-Up *p*-Value |
|---|---|---|---|---|---|---|---|---|---|---|
| Phosphorus | mg kg$^{-1}$ | 5.46 | 9.34 | 4.56 | 10.14 | 4.15 | 7.14 | * 0.01 | * 0.01 | * 0.01 |
| Potassium | mg kg$^{-1}$ | 39.92 | 46.56 | 58.61 | 53.20 | 34.35 | 43.51 | 0.34 | 0.50 | 0.16 |
| Calcium | mg kg$^{-1}$ | 406.29 | 430.16 | 388.10 | 445.43 | 246.83 | 438.03 | 0.73 | 0.37 | * 0.03 |
| Magnesium | mg kg$^{-1}$ | 57.02 | 69.71 | 63.08 | 76.89 | 44.37 | 67.49 | 0.15 | 0.13 | * 0.02 |
| Sulfur | mg kg$^{-1}$ | 4.48 | 4.84 | 4.55 | 4.77 | 3.49 | 4.02 | 0.47 | 0.66 | 0.21 |
| Sodium | mg kg$^{-1}$ | 53.97 | 60.88 | 60.71 | 58.45 | 61.14 | 55.94 | * 0.01 | 0.39 | * 0.01 |
| pH Water | | 4.50 | 5.17 | 4.77 | 5.08 | 5.08 | 5.58 | 0.98 | 0.73 | 0.86 |
| Estimated CEC | cmoles kg$^{-1}$ | 5.98 | 5.28 | 7.77 | 5.61 | 7.41 | 5.91 | 0.38 | * 0.01 | * 0.04 |
| Electrical Conductivity | µS cm$^{-1}$ | 71.51 | 74.59 | 44.24 | 84.53 | 34.08 | 43.49 | 0.84 | * 0.02 | * 0.10 |
| Carbon/Nitrogen Ratio | | 23.13 | 20.16 | 24.10 | 20.73 | 22.06 | 19.10 | * 0.01 | * 0.02 | * 0.04 |
| Organic Matter | % | 0.04 | 0.03 | 0.05 | 0.03 | 0.04 | 0.03 | * 0.10 | * 0.01 | 0.24 |
| Total Carbon | % | 2.01 | 1.59 | 2.36 | 1.58 | 1.80 | 1.46 | * 0.10 | * 0.01 | 0.24 |
| Total Nitrogen | % | 0.09 | 0.08 | 0.10 | 0.08 | 0.08 | 0.08 | 0.47 | * 0.08 | 0.72 |
| Nitrogen as Ammonium | mg kg$^{-1}$ | 5.87 | 4.81 | 10.40 | 7.74 | 5.14 | 5.07 | 0.32 | 0.43 | 0.96 |
| Nitrogen as Nitrate | mg kg$^{-1}$ | 2.86 | 9.35 | 15.96 | 8.72 | 2.17 | 2.10 | 0.32 | 0.14 | 0.96 |

**Table 3.** Soil nitrogen as ammonium, nitrate, total nitrogen and carbon/nitrogen (C/N) ratios. Pre = Pre-burn; Post = Post-burn; Green = Green-up.

| Subplot | NH4 mg kg$^{-1}$ Pre | Post | Green | Nitrate mg kg$^{-1}$ Pre | Post | Green | Total N % Pre | Post | Green | C/N Ratio Pre | Post | Green |
|---|---|---|---|---|---|---|---|---|---|---|---|---|
| 67.02-2 | 8.5 | 2.7 | 5.1 | 0.2 | 8.7 | 4.0 | 0.12 | 0.12 | 0.17 | 21.9 | 19.5 | 22.1 |
| 67.02-1 | 7.3 | 3.3 | 5.0 | -- | 4.3 | 0.7 | 0.10 | 0.08 | 0.09 | 23.2 | 20.6 | 23.9 |
| 66.02-2 | 4.5 | 3.3 | 3.1 | 0.3 | -- | 0.3 | 0.06 | 0.12 | 0.06 | 21.4 | 21.4 | 20.6 |
| 66.02 | 7.2 | 13.9 | 3.6 | 5.2 | 18.3 | -- | 0.13 | 0.09 | 0.05 | 23.1 | 20.9 | 22.2 |
| 66.01-2 | 4.0 | 4.3 | 2.3 | -- | -- | -- | 0.05 | 0.13 | 0.09 | 24.7 | 26.0 | 20.4 |
| 66.01 | 4.2 | 35.7 | 17.7 | 0.6 | 55.8 | 3.7 | 0.06 | 0.15 | 0.06 | 25.7 | 28.4 | 18.6 |
| 19.01-2 | 8.1 | 3.6 | 2.3 | 3.8 | 8.0 | -- | 0.09 | 0.04 | 0.08 | 21.3 | 32.0 | 25.7 |
| 19.01 | 3.1 | 16.5 | 2.4 | 7.2 | 0.6 | -- | 0.09 | 0.06 | 0.04 | 23.8 | 23.9 | 23.0 |
| W8-1 | 4.7 | 21.6 | 3.3 | 0.8 | 14.3 | 5.6 | 0.10 | 0.08 | 0.05 | 23.3 | 22.3 | 23.3 |
| W8 | 7.3 | 2.3 | 5.3 | 3.7 | 3.5 | 0.6 | 0.08 | 0.06 | 0.10 | 21.8 | 19.8 | 27.4 |

**Table 3.** *Cont.*

| Subplot | NH4 mg kg$^{-1}$ Pre | Post | Green | Nitrate mg kg$^{-1}$ Pre | Post | Green | Total N % Pre | Post | Green | C/N Ratio Pre | Post | Green |
|---|---|---|---|---|---|---|---|---|---|---|---|---|
| W7-1 | 2.8 | 5.2 | 6.2 | 0.8 | 3.5 | 0.8 | 0.05 | 0.02 | 0.03 | 27.3 | 22.1 | 20.1 |
| W7 | 15.1 | 31.1 | 2.3 | 71.1 | 16.7 | -- | 0.09 | 0.05 | 0.16 | 22.2 | 24.0 | 17.6 |
| W17-1 | 2.3 | 12.8 | 3.5 | -- | 13.7 | 1.0 | 0.09 | 0.09 | 0.07 | 23.4 | 23.7 | 16.3 |
| W17 | 2.9 | 10.3 | 1.8 | 0.3 | 9.6 | 5.6 | 0.08 | 0.12 | 0.10 | 18.3 | 21.2 | 19.9 |
| W16-1 | 3.4 | 5.2 | 2.9 | 0.1 | 6.5 | 7.7 | 0.11 | 0.07 | 0.06 | 21.4 | 18.0 | 16.7 |
| W16 | 4.8 | 3.0 | 5.2 | 0.6 | 3.9 | 0.8 | 0.14 | 0.06 | 0.07 | 21.8 | 19.1 | 19.6 |
| W15-1 | 3.8 | 3.4 | 4.8 | 0.1 | 4.8 | 0.5 | 0.06 | 0.07 | 0.03 | 23.5 | 18.8 | 15.0 |
| W15 | 2.9 | 2.8 | 6.9 | 5.4 | 5.6 | 2.4 | 0.06 | 0.06 | 0.04 | 18.6 | 21.8 | 14.4 |
| W14-1 | 4.3 | 2.8 | 6.5 | 0.7 | 3.9 | -- | 0.14 | 0.08 | 0.05 | 19.5 | 18.4 | 12.2 |
| W14 | 5.2 | 11.0 | 5.4 | 1.1 | 13.9 | -- | 0.06 | 0.05 | 0.07 | 20.2 | 18.8 | 15.5 |
| W13-1 | 2.7 | 19.1 | 3.7 | 1.0 | 1.6 | 1.4 | 0.06 | 0.09 | 0.13 | 22.2 | 17.2 | 17.9 |
| W13 | 3.1 | 3.7 | 3.1 | 1.0 | 12.9 | 0.3 | 0.11 | 0.12 | 0.06 | 18.8 | 17.3 | 13.6 |
| W12-1 | 5.6 | 24.0 | 3.0 | 6.0 | 17.9 | -- | 0.07 | 0.07 | 0.14 | 19.4 | 20.0 | 14.9 |
| W12 | 3.2 | 8.0 | 2.4 | 1.0 | 0.5 | -- | 0.09 | 0.09 | 0.08 | 17.8 | 17.3 | 13.5 |
| W11-1 | 2.2 | 10.2 | 2.3 | 0.1 | 22.5 | -- | 0.04 | 0.08 | 0.09 | 19.4 | 21.1 | 24.3 |
| W11 | 2.2 | 10.9 | 3.0 | 1.0 | 2.1 | -- | 0.05 | 0.10 | 0.11 | 15.9 | 20.9 | 23.9 |
| W05.1-1 | 2.6 | 3.0 | 3.3 | 17.8 | 4.7 | -- | 0.07 | 0.10 | 0.05 | 21.5 | 16.4 | 18.8 |
| W05.1 | 6.0 | 6.5 | 1.8 | 5.4 | 13.2 | -- | 0.05 | 0.03 | 0.05 | 20.9 | 29.3 | 21.4 |
| W05-1 | 9.3 | 2.8 | 2.4 | 4.7 | 3.4 | 0.6 | 0.13 | 0.07 | 0.09 | 19.9 | 21.2 | 18.1 |
| W05 | 2.3 | 5.5 | 5.2 | 13.0 | 21.6 | 3.0 | 0.05 | 0.06 | 0.04 | 18.5 | 18.9 | 23.3 |
| W04-1 | 6.6 | 3.5 | 8.9 | 7.4 | -- | 0.3 | 0.09 | 0.10 | 0.08 | 17.7 | 18.4 | 19.3 |
| W04 | 4.5 | 3.9 | 6.8 | 15.7 | -- | -- | 0.04 | 0.08 | 0.07 | 16.9 | 19.3 | 19.4 |
| W03-1 | 7.1 | 2.2 | 5.8 | 47.1 | -- | 0.2 | 0.09 | 0.07 | 0.05 | 18.9 | 20.7 | 26.0 |
| W03 | 2.5 | 1.8 | 5.9 | 14.0 | -- | 0.1 | 0.05 | 0.08 | 0.08 | 21.2 | 26.1 | 20.0 |
| W02-1 | 8.6 | 2.6 | 6.9 | 20.6 | -- | -- | 0.08 | 0.06 | 0.10 | 16.1 | 20.0 | 20.00 |
| W02 | 5.3 | 2.8 | 6.4 | 5.8 | -- | -- | 0.11 | 0.14 | 0.09 | 18.5 | 19.7 | 18.8 |
| W01-1 | 6.0 | 5.4 | 18.0 | 20.1 | 2.9 | 3.9 | 0.06 | 0.06 | 0.05 | 16.6 | 28.5 | 21.0 |
| W01 | 5.1 | 4.8 | 9.2 | 5.1 | 5.9 | 3.0 | 0.06 | 0.13 | 0.07 | 23.6 | 21.8 | 21.1 |

## 4. Discussion

The significant changes in the extractable phosphorus concentration in our study were inconsistent with other findings [5,7], who reported no difference in phosphorus after a burn. Even though we detected a significant decline in the available phosphorus, the concentrations of phosphorus in the soil were relatively low across the different times and not at a more available level of concentration in the soil [18]. Significant changes in potassium agreed with [5], who found a significant increase in potassium after a fire, but contradicts [7], who found no significant change in potassium after a burn. Even though there were significant differences between time frames and burn intervals, they were not enough to effectively change the level of potassium [18]. Calcium was only significantly different between the two burn intervals at the green-up time frame, which is inconsistent with [5,7], who found calcium to increase after a fire. Calcium was at an optimum level during the pre-burn and post-burn time frames and was slightly below optimum at the green-up time frame, so there may have been a slight influence from a prescribed fire in our study [18]. The significant increase in magnesium from the pre-burn to post-burn time frame followed by a significant decrease from the post-burn to green-up time frame was inconsistent with [5], who reported a significant decrease after a burn. Since magnesium levels were high throughout this study, there was no influence of prescribed fire on the levels of magnesium [18]. Sulfur in our study significantly decreased over time; since sulfur easily volatilizes during a fire, there are often decreases in sulfur directly post-fire, so these results agree with [19] but contradict [5], who did not find any differences in sulfur after the burn. Sulfur levels throughout this study were low, indicating there was not enough to change the category level of sulfur [18]. Significant changes in sodium in our study agree with [19]; however, sodium levels were low throughout the study, indicating that prescribed fires had little effect on sodium in the soil [20].

The soil pH significantly increased directly after a fire, especially after a hotter burn [7,9,13,14]. The pH showed a decrease throughout all time frames and significantly decreased over time, contradicting previous studies [7,9,13,14]. The CEC increased over time, contrasting with [21], who reported a decrease after a fire.

Higher electrical conductivity indicates higher concentrations of ions in the soil solution. The electrical conductivity significantly decreased from post-burn to green-up and from pre-burn to green-up, and there was also a significant difference between the two different burn intervals at the post-burn and green-up time frames. This indicates a decrease in ions in the solution and contradicts [21] who found electrical conductivity increased after a fire.

The carbon-to-nitrogen ratio had a significant decrease from post-burn to green-up. It was also significantly different between the two different burn intervals at all three intervals. This agrees with [22], who reported a decrease from the unburned plots to the burned plots. The soil organic matter percentage was only significantly different between the two burn intervals at the pre-burn and post-burn time frames, and there was no significant difference in the soil organic matter percentage, total carbon, and total nitrogen over time. The percentage of total nitrogen was significant between the two burn intervals at the post-burn time frame. The total carbon percentage was significant between the two burn intervals at the pre-burn and post-burn time frames. Even though there was not a significant change over time, the slight changes were enough to significantly change the carbon-to-nitrogen ratio.

Both ammonium and nitrate increased significantly after a prescribed burn and did not decrease for some time afterwards [23]; our results partially contradict [23], who found an increase in both ammonium and nitrate, with nitrogen as ammonium initially increasing significantly, followed by both significantly decreasing. This may be caused by the soil moisture content and by the leaching of nitrogen further into the soil profile. Another possibility is the sandy soil texture, which holds less water and leaches more easily [24].

## 5. Conclusions

While prescribed burns may help in the reduction of the O-Horizon, the O-Horizon density did not follow trends normally associated with the O-Horizon after a fire. Typically, there is a significant decrease followed by a slow buildup and this study found the opposite. Fuel loads were not measured during this study but may have had an effect on soil physical and chemical properties such as soil water infiltration rates and nutrient availability. Prescribed burns have been used to decrease the likelihood of a wildfire and increase some nutrients. The total carbon percentage within the soil organic matter showed slight changes over time, mostly decreasing. This indicates that prescribed burning does not have the negative impact of releasing soil carbon into the atmosphere; the carbon is still retained within the soil. Given the global interest in increased carbon sequestration, the study eases concerns about the release of carbon by fire into the atmosphere. Further directed studies are needed to determine the amount of carbon contained in the soil against how much is released. But the initial data suggest land managers should continue using prescribed burns as a habitat management tool without fear of increasing carbon into the atmosphere.

**Supplementary Materials:** The following supporting information can be downloaded at: https://www.mdpi.com/article/10.3390/f14091912/s1, Table S1. Site location, subplot code, and the GPS coordinates for each sampling plot. Winston 8 = Winston 8 Land and Cattle Tree Farm; Angelina NF = Angelina National Forest; Davy Crockett NF = Davy Crockett National Forest; Table S2. Confirmed or corrected soil map unit and upper 15 cm soil texture. Winston 8 = Winston 8 Land and Cattle Tree Farm; ANF = Angelina National Forest; DC NF = Davy Crockett National Forest; Table S3. Phosphorus (P), Potassium (K), Calcium (Ca), Magnesium (Mg) and Sulfur (S) levels. Pre = Pre-burn; Post = Post burn; Green = Green-up; Table S4. Soil pH, Sodium (Na), Cation Exchange capacity (CEC), Electrical conductivity levels. Pre = Pre-burn; Post = Post burn; Green = Green-up.

**Author Contributions:** C.P.D., B.P.O. and K.W.F. contributed to the study design and manuscript preparation. C.P.D. was responsible for all of the field data collection and data analysis. All authors have read and agreed to the published version of the manuscript.

**Funding:** This project was funded by the Division of Environmental Science, Arthur Temple College of Forestry and Agriculture at Stephen F. Austin State University.

**Data Availability Statement:** The datasets used for this study are available from the corresponding author upon request.

**Acknowledgments:** Access to research plots was graciously provided by the National Forests and Grasslands of Texas of the United States Forest Service and the Winston 8 Land and Cattle Ltd. Tree Farm. We sincerely thank Cole Dunson, William Steinley, Wesley Danheim, Wyatt Bagwell, Hanna Haydon, and Lauren Lara for data collection assistance on this project.

**Conflicts of Interest:** The authors declare they have no conflicts of interest.

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
