# Peer review of "Comparing the Effects of Prescribed Burning on Soil Chemical Properties in East Texas Forests with Longleaf and Other Southern Pines in the Overstory"

_forests, doi:10.3390/f14091912_

Round 1

Reviewer 1 Report

Overall comments:

Besides Table 6, which lists National Forests as having one burn interval, and Winston 8 having another burn interval, I did not find in the text which sites had the 2–3 fire return interval versus the 2-year fire return interval. Lines 135–145 of the Results section mention some differences between “burn intervals” (I assume this means sites) and refers to Table 4, which does not have information on sites. I am wondering if there were differences in soil chemical properties by sites, how did the properties vary by sites (without the reader having to make sense of the raw data in Tables 3 and 5). And if there were differences among sites could this be due to the fire return interval, tree species composition, or both. Surprising none of this is mentioned in the Discussion, especially because species composition is mentioned in the Highlights and Abstract.    

For the highlights consider the following revision: “Results from this study in east Texas suggest that cover types dominated by longleaf pine had little overall impact on soil chemical properties compared to cover types dominated by shortleaf pine or loblolly pine. However, there were some short-term changes. Regardless of overstory composition, the fire return interval had a significant influence on some chemical properties of these sandy and sandy loams soils.”

Line-by-line comments (Introduction–Results):

Line 26: Replace “Results found” with “We found”.

Lines 27–29: It might be helpful to indicate whether there were increases or decreases in these soil properties.

Line 36: Instead of just “on soil”, add “on soil properties”.

Tables and Figures:

Remove figure 1. The description of the plot layout in the text is adequate.

Table 1 should be included in the supplemental materials. The reader does not need a list of plot coordinates to understand the research findings.

Table 2 should be included in the supplemental materials. It is sufficient enough to list the different soil series and textures in the methods or results.

Tables 3, 5, and 7 should be included in the supplemental materials. These tables simply show the values for each subplot by time period. The reader cannot easily make conclusions based on tables of the raw data.

Table 4. Should the p-value in the caption be 0.1 instead of 0.01? In the methods section it says 0.1. I like the way the results are presented in this table.

Line-by-line comments (Discussion–Conclusion):

Discussion, line 3: It is important to remind readers that this is “extractable phosphorus concentration”. Be sure that the cited studies also evaluated concentrations and not stocks. This way proper comparisons can be made. I did not check the other studies to confirm whether concentrations or stocks were evaluated.

Discussion, line 5: No effect? I would revise this sentence to something like: “Even though we detected a significant decline in available phosphorus, the concentrations of phosphorus in the soil were relatively low across time.”

Discussion, line 8: The authors need to be stated here, not just [5]. Same with [7] in line 9, and so on.

Discussion, lines 3–27: Besides line 21 about S and volatilization there is little discussion on why the changes in soil chemical properties occurred or why the results were different than those of other studies. This is true of the rest of the discussion, too.

Conclusion, line 63: I think you need to specifically say “releasing soil carbon into the atmosphere” because you did not measure all ecosystem stocks. Furthermore, you evaluated models of % C (concentration) and not the C stock or C mass, so you cannot really say whether or not C was released to the atmosphere. You would need to know the soil mass or the bulk density for the 0 to 15 cm depth increment to calculate the C stock or mass.

Author Response

Our responses are below each comment and highlited in yellow.

Besides Table 6, which lists National Forests as having one burn interval, and Winston 8 having another burn interval, I did not find in the text which sites had the 2–3 fire return interval versus the 2-year fire return interval. Lines 135–145 of the Results section mention some differences between “burn intervals” (I assume this means sites) and refers to Table 4, which does not have information on sites. I am wondering if there were differences in soil chemical properties by sites, how did the properties vary by sites (without the reader having to make sense of the raw data in Tables 3 and 5). And if there were differences among sites could this be due to the fire return interval, tree species composition, or both. Surprising none of this is mentioned in the Discussion, especially because species composition is mentioned in the Highlights and Abstract.   

We attempted to clarify in the text the burn interval and sites.

Table 4 citation is in (3,4,5) to cover all info in that paragraph, rather than pull out each table in different lines in that paragraph.

For the highlights consider the following revision: “Results from this study in east Texas suggest that cover types dominated by longleaf pine had little overall impact on soil chemical properties compared to cover types dominated by shortleaf pine or loblolly pine. However, there were some short-term changes. Regardless of overstory composition, the fire return interval had a significant influence on some chemical properties of these sandy and sandy loams soils.”

We revised as suggested

Line-by-line comments (Introduction–Results):

Line 26: Replace “Results found” with “We found”.

Done

Lines 27–29: It might be helpful to indicate whether there were increases or decreases in these soil properties.

Done

Line 36: Instead of just “on soil”, add “on soil properties”.

Done

Tables and Figures:

Remove figure 1. The description of the plot layout in the text is adequate.

Done

Table 1 should be included in the supplemental materials. The reader does not need a list of plot coordinates to understand the research findings.

Table 2 should be included in the supplemental materials. It is sufficient enough to list the different soil series and textures in the methods or results.

Tables 3, 5, and 7 should be included in the supplemental materials. These tables simply show the values for each subplot by time period. The reader cannot easily make conclusions based on tables of the raw data.

Done for Tables 1,2,3,5,7 All are renumbered.

Table 4. Should the p-value in the caption be 0.1 instead of 0.01? In the methods section it says 0.1. I like the way the results are presented in this table.

Done

Line-by-line comments (Discussion–Conclusion):

Discussion, line 3: It is important to remind readers that this is “extractable phosphorus concentration”. Be sure that the cited studies also evaluated concentrations and not stocks. This way proper comparisons can be made. I did not check the other studies to confirm whether concentrations or stocks were evaluated.

Done

Discussion, line 5: No effect? I would revise this sentence to something like: “Even though we detected a significant decline in available phosphorus, the concentrations of phosphorus in the soil were relatively low across time.”

Done

Discussion, line 8: The authors need to be stated here, not just [5]. Same with [7] in line 9, and so on.

We haven’t seen it done this way in this journal

Discussion, lines 3–27: Besides line 21 about S and volatilization there is little discussion on why the changes in soil chemical properties occurred or why the results were different than those of other studies. This is true of the rest of the discussion, too.

We think this has now been addressed,

Conclusion, line 63: I think you need to specifically say “releasing soil carbon into the atmosphere” because you did not measure all ecosystem stocks. Furthermore, you evaluated models of % C (concentration) and not the C stock or C mass, so you cannot really say whether or not C was released to the atmosphere. You would need to know the soil mass or the bulk density for the 0 to 15 cm depth increment to calculate the C stock or mass.

Done

Reviewer 2 Report

The research wants to study the correlations between the soil chemical properties among differing burn intervals and the effects prescribed burning has on them. The research should be interested to readers, but there are many flaws in the paper and need to improve. There are no details about the experiment, for example, the strong level of the prescribed fire, soil sampling methods, measuring methods of soil chemical properties. The plot choosing is too random, 30 in one site and only 2 in another. The tables in the results are not good to read, and figs would be better. The discussion and introduction are too shallow and no relation with the conclusion. The references are not enough.

The English writing is fluent and understandable.  

Author Response

Reviewer 2

The research wants to study the correlations between the soil chemical properties among differing burn intervals and the effects prescribed burning has on them. The research should be interested to readers, but there are many flaws in the paper and need to improve. There are no details about the experiment, for example, the strong level of the prescribed fire, soil sampling methods, measuring methods of soil chemical properties. The plot choosing is too random, 30 in one site and only 2 in another. The tables in the results are not good to read, and figs would be better. The discussion and introduction are too shallow and no relation with the conclusion. The references are not enough.

Our response:

We disagree that there are no details about the experiment (see sections 2.1, 2.2). The only parameters regarding fire we had available was fire intervals, not fire behavior.  Fire behavior was not a part of the study.  There were not 30 plots from 1 site, 2 from another, but 8, 2, 30).  Plot placement were applied randomly.  We acknowledge that the study had an unbalanced design, but it was random, and we analyzed the data knowing it was unbalanced.

There were too many variables presented in the tables to have been placed in figures in our opinion.

We are not sure what reviewer 2 means regarding “too shallow” and “not enough” in the discussion and references.  We know there are many more articles out there, but not that many for the west gulf coast (which we included) and just adding references that state the same thing is not useful.   We also disagree that the introduction and discussion are not connected to the conclusions.